# Smart Farming through Responsible Leadership in Bangladesh: Possibilities, Opportunities, and Beyond

Amlan Haque [1,*], Nahina Islam [2,3], Nahidul Hoque Samrat [4], Shuvashis Dey [5] and Biplob Ray [2,3]

1 School of Business & Law, CQUniversity, Sydney 2000, Australia
2 Centre for Intelligent Systems, School of Engineering and Technology, CQUniversity, Rockhampton 4700, Australia; n.islam@cqu.edu.au (N.I.); b.ray@cqu.edu.au (B.R.)
3 Institute for Future Farming Systems, CQUniversity, Bundaberg 4670, Australia
4 School of Health, Medical and Applied Sciences, CQUniversity, Bundaberg 4670, Australia; n.samrat@cqu.edu.au
5 Department of Electrical & Computer Systems Engineering, Monash University, Clayton 3800, Australia; shuvashis.dey@monash.edu
* Correspondence: a.haque@cqu.edu.au

**Abstract:** Smart farming has the potential to overcome the challenge of 2050 to feed 10 billion people. Both artificial intelligence (AI) and the internet of things (IoT) have become critical prerequisites to smart farming due to their high interoperability, sensors, and cutting-edge technologies. Extending the role of responsible leadership, this paper proposes an AI and IoT based smart farming system in Bangladesh. With a comprehensive literature review, this paper counsels the need to go beyond the simple application of traditional farming and irrigation practices and recommends implementing smart farming enabling responsible leadership to uphold sustainable agriculture. It contributes to the current literature of smart farming in several ways. First, this paper helps to understand the prospect and challenges of both AI and IoT and the requirement of smart farming in a nonwestern context. Second, it clarifies the interventions of responsible leadership into Bangladesh's agriculture sector and justifies the demand for sustainable smart farming. Third, this paper is a step forward to explore future empirical studies for the effective and efficient use of AI and IoT to adopt smart farming. Finally, this paper will help policymakers to take responsible initiatives to plan and apply smart farming in a developing economy like Bangladesh.

**Keywords:** sustainable farming; artificial intelligence (AI); internet of things (IoT); smart farming; responsible leadership; remote communication





## 1. Introduction

The 2050 goal of sustainable farming is to increase agricultural yield to meet the food demand of 10 billion people and calls for smart farming [1]. The concept of smart farming refers to the application of cutting-edge technologies, including artificial intelligence (AI), internet of things (IoT), and high-speed wireless network into farming practices to maximize agricultural production and sustainability [2–4]. Following the trend and development, traditional agriculture practices are converting into new wireless and innovative technologies for both cultivation and supply chain management in Bangladesh. Moreover, the concept of Agri-Food 4.0 has initiated a revolution in traditional farming by introducing the fifth generation or 5G wireless communication technology to the field [4]. Agri-Food 4.0 refers to a comprehensive digitalized approach for higher productivity and practical and efficient land management to achieve agricultural sustainability, quality of life for farmers, and overall competitiveness of the sector [5].

On the other hand, the role of leadership has been identified as one of the significant contributors to agricultural development across the world [6]. The use of AI, IoT, wireless network, and responsible leadership in the agriculture sector in Bangladesh may install

and uphold both the effectiveness and efficiency of smart farming. The concept of responsible leadership has moved into the leadership literature as a multilevel phenomenon and defined as a "values-based and principle-driven relationship between leaders and stakeholders who are connected through a shared sense of meaning and purpose through which they raise to higher levels of motivation and commitment for achieving sustainable value creation and responsible change (p. 539)" [7]. Responsible leadership with this combined approach has the potential to improve the existing cultivation practices, quality improvement, reduction of production cost, and opportunity to mitigate both the socioeconomic and environmental footprints of traditional farming [8]. As a result, smart farming can lead farmers and economies to the modern and intelligent system by monitoring and developing sustainable agriculture. In this paper, we develop a conceptual framework emphasizing the solutions for AI and IoT based smart farming and the role of responsible leadership for sustainable agriculture in the context of Bangladesh. The proposed framework contributes to the need for a detailed and contextualized understanding of AI and IoT based smart farming in Bangladesh in several ways.

First, the concept of AI and IoT based smart farming is an emerging concept, and the application of responsible leadership and its alignment with traditional agriculture was found to be challenging [2,3,9]. Responsible leadership increases performance outcomes and can support two significant imperatives for smart farming, such as a shared sense of meaning and purposes for achieving sustainable production [10,11]. For example, responsible leadership upholds a value-based relationship among leaders (e.g., policymakers) and stakeholders (e.g., farmers and consumers) for responsible change toward sustainable development of the agriculture sector [3,10,11]. This paper contributes to this understanding by integrating responsible leadership into smart farming and exploring AI and IoT based smart farming for sustainable agriculture in Bangladesh.

Second, several scholars suggested the prospect of IoT for smart farming [4,12]. Clarifying the concept of smart farming enabling responsible leadership, this paper contributes to the current literature of AI and IoT based agricultural studies and current understanding of how smart farming and responsible leadership can improve sustainable production in a nonwestern context.

Third, previous studies pointed out the need for responsible leaders to establish smart farming and secure sustainable production [13,14]. This paper clarifies the concept of responsible leadership and its suitability for smart farming in Bangladesh for sustainable performance.

Finally, the agriculture sector and the farmers play a vital role in food security and poverty reduction to strengthen Bangladesh's economic condition. Scholars suggested that traditional farming is predominant in most remote village areas in Bangladesh and the achievement of the United Nations' 2030 Sustainable Development Goal (SDG: 2.4.1) for sustainable agriculture appears to be very challenging [15]. This paper is a step forward to not only explore the future research avenues for the opportunity of smart farming enabling responsible leadership in Bangladesh but will also help the policymakers to take responsible initiatives to achieve 2030 sustainable agriculture. Sections 2–8 of this paper present the current communication and agricultural technology scenario in Bangladesh (Section 2), followed by the potential of smart farming in Section 3. We propose an AI and IoT based autonomous smart irrigation system in Section 4. Accordingly, the fundamental concept of responsible leadership and its role are described in Sections 5 and 6. Finally, we summarize future research directions in Section 7 and present our conclusions in Section 8.

## 2. Present Status of Communication and Agricultural Technology in Bangladesh

### 2.1. Wireless Communication

The mobile telecom industry in Bangladesh is expanding rapidly and making a significant contribution to the economy of Bangladesh. With the development of technology, the telecom sector has advanced a lot. Bangladesh was introduced with wireless telephony just before the beginning of the 1990s. Since then, the mobile telecom sector has changed

significantly. In the past, people used mobile phones for voice communication or text messages only, but with the emergence of smart technologies, people are now using data and content-based services. At present, most internet users in Bangladesh have smartphones to get access to the internet for their daily life purposes such as mobile banking, social networking, and streaming services.

The telecommunication companies in Bangladesh started providing 4G network in 2018, which was a crucial step towards fulfilling Bangladesh government's aim to make "Digital Bangladesh". There are four major telecommunication service providers in Bangladesh: Grameen Phone (GP), Banglalink, Robi and Teletalk. All these operators provide 4G network coverage throughout Bangladesh and support a total of 158.438 million mobile subscribers [16,17], among which 86.268 million subscribers use mobile internet. They aim to launch 5G network and services by 2021 [16]. Hence, it is evident that Bangladesh has good network coverage and connectivity throughout the country, which ensures the potentiality of smart farming in Bangladesh.

To understand the adaptability of farmers towards smart technologies and the status of the communication network in Bangladesh, we analyzed the data gathered from a survey, which was conducted in Bangladesh by Jannat et al. [18]. In their survey, approximately 125 participants were asked about what type of phone they use, and 121 (96.8%) of them were found to have smartphones. On the other hand, only two persons said that they use feature phones. The survey results also revealed the increasing popularity of smartphones, even among lower income-holders due to its reasonable price. This is because of different features available in smartphones, such as a high-resolution camera, internet accessibility, and easy accessibility to social networking sites, banking, emailing etc. In the same survey [18], participants were asked about which technology is available in the mobile network they use, around 104 respondents were using 4G technology, and the other 21 respondents use 3G technology. Hence, it was found that 4G network is accessible to 83.2% users, whereas 16.8% uses the 3G network. The percentage of using 4G was 83.2%, and 16.8% were 3G users. The participants were also asked about the speed of the internet, such as streaming services speed, picture uploading speed in social media etc. The feedback from the respondents was quite impressive, as more than 80% of participants confirmed that the internet speed is faster than it used to be with 3G. Moreover, the respondents seem to be happy with the video clarity and streaming services, as the 4G technology ensures higher bandwidth and high-quality videos. In addition, 117 (98.3%) among 125 respondents advised that they use apps for video calling purposes. This is due to the popularity of Messenger, Viber and WhatsApp to communicate with others from around the globe via video calls. These applications are facilitated by the high speed of internet connection provided by Wi-Fi or cellular technologies. All the developments mentioned above have influenced the mobile phone culture in the village areas for the farmers and hence, have opened the door for establishing smart farming in Bangladesh.

*2.2. Building Low-Power Wide Area Networks (LPWANs) Where Cellular Network Connectivity Is Limited*

The cellular network in some remote villages (such as in the southeast corner of Bangladesh) is not strong enough to support AI and IoT based farming in Bangladesh. For this situation, the low-power wide area networks (LPWAN) is an alternative, cost-effective communication technology to support AI and IoT applications [19,20]. A widely used LPWAN standard is LoRaWAN, which was developed by LoRa Alliance and has key features such as long-range communication, low energy consumption, GPS-free positioning, built-in security, etc. The researchers proposed LoRaWAN as a better solution for smart farming, particularly for remote areas as it can cover a long range and consumes significantly less energy [20,21]. They also suggested that the LoRaWAN communication system can enable AI and IoT networks over 10 km, wirelessly in remote settings without any support of long-term evolution (LTE-4G/5G) or other backhaul networks. It also consumes as low energy as only 1536 mAh per day. LoRaWAN networks are also becoming popular in Bangladesh to provide connectivity for smart technologies. For instance, an air

quality monitoring system in Bangladesh, using mq-2, mq-3 and mq-7 sensors attached to a microcontroller is connected to the internet-supported LoRaWAN network [12]. Moreover, LoRaWAN setup in Bangladeshi public service areas (e.g., healthcare) is also found to be fruitful in achieving activity recognition and monitoring via LoRaWAN sensors [12,16].

*2.3. Agriculture Technology and System*

Bangladesh's agricultural sector is the primary source of livelihood for rural people. The sector has changed significantly over the past few decades and contributes 14.79% to the total gross domestic product (GDP) of the country. Currently, about 45.1% of the labor force is accommodated by this sector [22]. As one of the most densely populated countries, overall agricultural productivity in Bangladesh and farmer's profit is crucial to reduce poverty in the rural development, employment generation, and future food security as the population is estimated to be 254,100,000 by 2050 [22]. As a result, Bangladesh needs more strategic support and responsible leadership to increase overall agricultural productivity to alleviate poverty by AI and IoT based smart farming. However, technology adoption in the agriculture sector is slow and steady among the farmers, and the widespread adoption of modern technologies is still needed to be achieved. Since the last two decades, several technologies are being used in the agriculture sector related to horticulture, agronomy, livestock, farming tools and machines. Sections 2.3.1–2.3.4 summarize the overall status of the Bangladeshi agriculture system to clarify our proposed AI and IoT-based smart irrigation and farming system.

2.3.1. Irrigation System

The water use for irrigation (groundwater, surface water, or rainwater) is highly inefficient in Bangladesh. Due to the faulty and old-fashioned irrigation systems, only about 25% to 30% of the water is used for irrigation [23]. Most farmers use groundwater from the surface level water. They use shallow tube wells with an electric pump. The irrigation cost and the overall production cost subsequently increase as around 80% as the irrigation pumps are diesel operated. Moreover, this diesel pump-based irrigation system contributes to greenhouse gas emissions from the country's agriculture sector. In addition, high levels of arsenic presence are suspected in the groundwater in northern and northwestern parts of Bangladesh [23]. However, we are yet to see any prospect for sensors or measurement equipment to measure arsenic level before irrigation. Thus, the inefficient irrigation is visible, and monsoon rainwater will not be the solution due to the lack of water storage facility, coordination, and responsible leadership.

In some parts of Bangladesh, an alternative wetting and drying (AWD) technology has been introduced as a water-efficient irrigation solution [24,25]. In AWD, a tube/pipe made of PVC is typically used to monitor the water depth and measure water availability in the field below the soil surface. The standard practice is to use a 30 cm long pipe with a diameter of 7–10 cm, which has perforations at the bottom part. Around 10 cm nonperforated portion stays above the surface, while the bottom of the perforated portion remains underneath the soil surface. The perforations allow the water to come inside the tube from the soil, where a scale measures water depth below the soil surface. Trials have shown that the AWD method saves about 365 mm of irrigation water (about 27%) over traditional irrigation practices [24,25]. Hence, AWD facilitates the optimal use of water for irrigation. However, this process is manual and labor intensive, therefore, indicating a further prospect of smart farming in Bangladesh.

2.3.2. Disease and Pests

Useful and timely plant disease diagnosis, pest control and prevention are critical for raising productivity. Scholars reported that around 4–14% of rice yield in Bangladesh is damaged each year because of different insect pests (e.g., nematode and bacterial leaf blight are the two common severe diseases in rice) [23]. The technology to automatically detect and map plant disease is very limited in Bangladesh. In most cases, farmers physically

scout to identify the plant disease and medicate them. This process is also found to be manual, time-consuming, and inefficient.

### 2.3.3. Pesticide and Fertilizer

Bangladeshi farmers have commonly used inorganic fertilizers (e.g., triple superphosphate, urea, muriate of potash) during the last four decades. Farmers are using inorganic fertilizer and overlooking their adverse effects on greenhouse gas emissions. Moreover, lack of knowledge and responsible leadership in the sector is not helping to promote the benefit and use of organic fertilizer. In most cases, farmers apply the inorganic fertilizer manually based on their assumption, which led to the disproportional use of fertilizer on the plants. They do not use any sensor to measure the soil–plant system's nutrient content before applying the fertilizers. This, in turn, led to increased nutrient imbalance, reduced production efficiency, and adverse effects on climate change and public health.

### 2.3.4. Agriculture Mechanization

In the last few decades, the pace of mechanization has accelerated in the agricultural sector in Bangladesh. Farmers use reapers to harvest and plant the next season crops mechanically. The two-wheel reaper is particularly prevalent in Bangladesh, especially among female growers, because it is easy to maneuver. The seed drills till, plant, and fertilize the crops by maintaining the sowing distance with greater precision and in lines simultaneously. These drills can allow farmers to plant using conservation agriculture practices like strip-tilling, a system that tills only small strips of land into which seed and fertilizer are placed. These drills are often used as attachments on two-wheeled tractors. This is a positive trend to investigate the prospect of smart farming in Bangladesh but requires responsible leadership for the overall sector.

## 3. Potential of Smart Farming in Bangladesh

Some problems of the agriculture sector in Bangladesh includes monocropping, loss of arable land, natural disaster and climate change, imbalanced use of pesticide and fertilizer, inefficient use of water, land degradation, inefficient ways of detecting of diseases and pest, and information and communication gaps (i.e., using of modern AI and IoT facilities) [26]. Over the last decade, new farming methods (known as smart farming) have been introduced with advancements in technology, and these methods have replaced the most used traditional farming methods. This approach includes aspects such as IoT, soil scanning, automatic irrigation, automatic plant disease detection, and data management. Over a few years, smart farming has become useful to all farmers (small and large scale). Now, there is no need for farmers to apply water, pesticides, and fertilizers evenly across the entire farm. Figure 1 shows the overall picture of how small and smart farming will be in the future.

Therefore, smart farming will allow farmers to use the lowest amounts of these elements and target specific areas of their farm. Below, we highlight some smart farming technology that can be applied in Bangladesh to increase product quality and quantity while reducing the farming cost.

### 3.1. Forecasting

Forecasting is one of the fundamental features of AI and IoT based smart farming that is used as a preventive measure that requires some actions due to a predicted event, for example, irrigating, weeding, or harvesting. In smart farming, the farms and crops are monitored and managed by accessing "real-time" data and historical data, and then using them to forecast several factors such as crop yield, productivity, etc. [27,28]. Hence, monitoring and documentation are essential prerequisites for enabling forecasting. As for forecasting tools, AI such as machine learning, deep learning, and scientific modeling is effective. Different machine learning models have been used in literature, for example,

artificial neural networks for forecasting maximum and minimum temperatures at field level [27,28], or forecasting soil moisture or plant disease detection, ref. [29] etc.

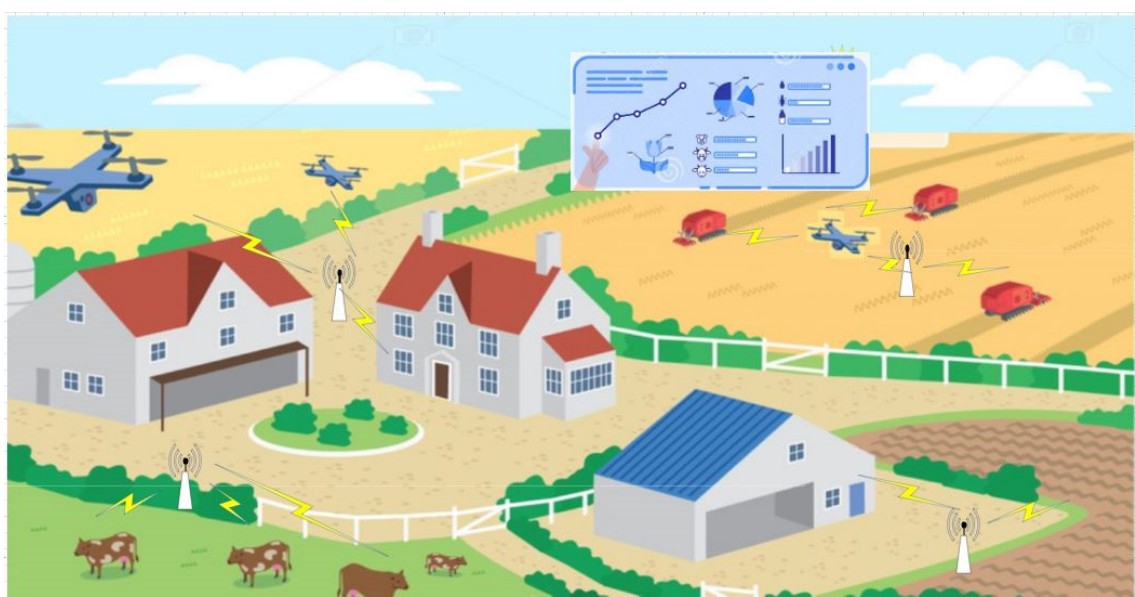

**Figure 1.** AI and IoT based future small and smart farming system. Modified with permission from https://www.nesta.org. uk/blog/precision-agriculture-almost-20-increase-in-income-possible-from-smart-farming/.

### 3.2. Remote Sensing

The most common smart farming application is to assess plant health by using remote sensing and image analytics. Remote sensing is an information collection process (through the image) of any object from a distance without making any physical contact with it. One of the most applied remote sensing methods is aerial monitoring by using images captured by satellites and unmanned and manned aircraft [19,20,25]. Often satellite imagery is very expensive for developing country farmers. Usually, the resolution and quality of satellite data are not practical and satisfactory due to the weather conditions. Accordingly, aerial data captured by manned aircraft present a better quality compared to the satellite data, but this method is also costly.

Conversely, small unmanned aerial vehicles (UAVs), also known as drones, can provide high-quality data more economically; Figure 2 shows the typical remote sensing platform used in smart farming [30]. Drones can collect imaging data over the ground through the high-resolution camera with corresponding geographic locations. It can carry high-resolution RGB, multispectral, hyperspectral, and thermal sensors or cameras to acquire the data [20,25]. Accordingly, these datasets can extract vegetation indices and temperature profiles that enable farmers to regularly monitor plant variability, health, and stress conditions.

One of the most useful sectors of smart farming is weed and infestation detection using UAVs. According to a study conducted in the United States, approximately 33 billion US$ of annual damage is caused by pest infestations and infections in crops [31]. Hence, early diagnosis is essential to minimize this damage. Many researchers have used RGB cameras, hyperspectral cameras, and multispectrum sensors mounted on UAVs to detect weed and infestation in plants and crops [29].

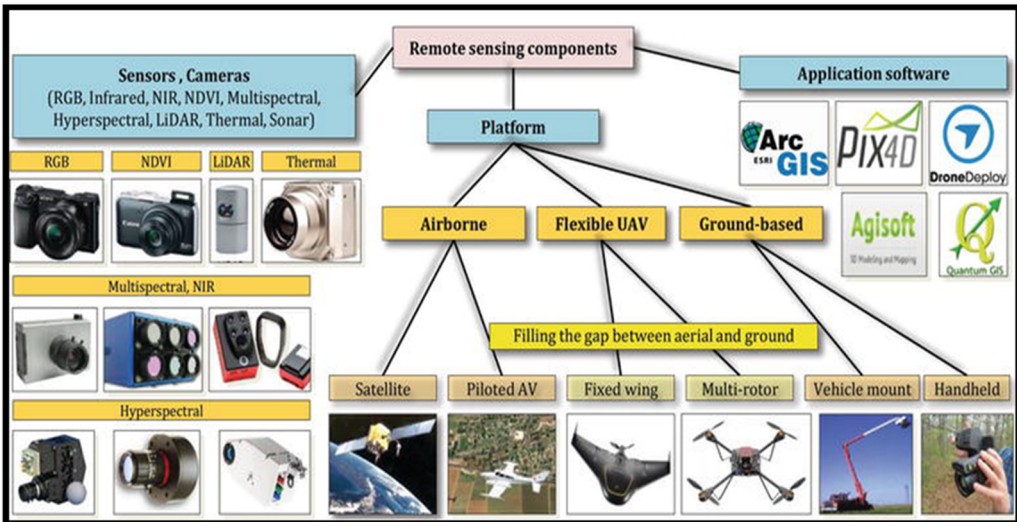

**Figure 2.** Typical components of a remote sensing platform for smart farming [30]. Copyright 2018 Licensee IntechOpen.

Drones are used to create useful 2D or 3D maps of an agricultural field. For example, the area of the farmland, soil conditions, and status of the crops and infestation within the crop can be detected in these maps [32]. For instance, researchers obtained high-resolution maps delineating the spatial variations of radiation interception using UAV images [33]. The maps generated allow for profitable smart farming tasks, such as the agronomic control of homogeneous zones and the separation of fruit quality areas.

Besides these processes, another possible use of drones in smart farming is spraying pesticides. The measure of pesticides per hectare of farmland correlates to the risks of worker ailments and environmental pollution. Spraying drones can carry large tanks whose capacity around 10 L, and it can spray one liter per minute, which means it is possible to spray one hectare within 10 min [34]. The drone-based spraying platform synchronized can be with an aerial crop monitoring process, thus providing accurate and efficient use of the agrochemical products. For example, researchers studied citrus farms to determine the optimum level of preventive work by spraying them from various heights using a UAV [35].

### 3.3. Planting Seeds and Seedlings

Planting seeds and seedlings can be made more efficient using UAVs. Authors in [35] presented the effective use of UAVs in a large area of uneven rice paddies. They used a UAV based system to distribute seeds and plant nutrients by accessing conditions for plant growth. Although using UAVs for planting seeds and seedlings is still in development, it is expected that this strategy will enhance efficiency, provided the UAV is equipped with image recognition technology and optimized planting tasks.

### 3.4. Sensors

A variety of sensors is used in smart farming to help farmers and agronomists make the best decisions regarding planting, fertilizing, and harvesting crops [36]. In Figure 3, we highlighted some of the most widely required sensors for site-specific crop management. The following subsections (Sections 3.4.1–3.4.4) highlight the essential sensors for our proposed AI and IoT based smart farming system and its outcome.

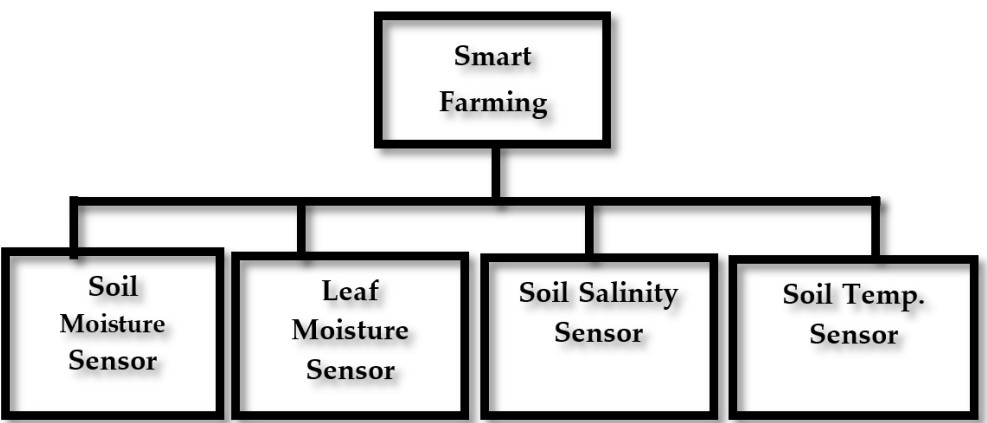

**Figure 3.** Sensors used in smart farming.

### 3.4.1. Soil Moisture Sensor

Knowledge of soil moisture contents at high resolution is quite important for effective irrigation scheduling and cropping prctices, improved weather and flood forecasting, and sustainable land and water management [37]. Water policy necessitates a complex spatial and temporal soil moisture measurement system, as shown in Figure 4.

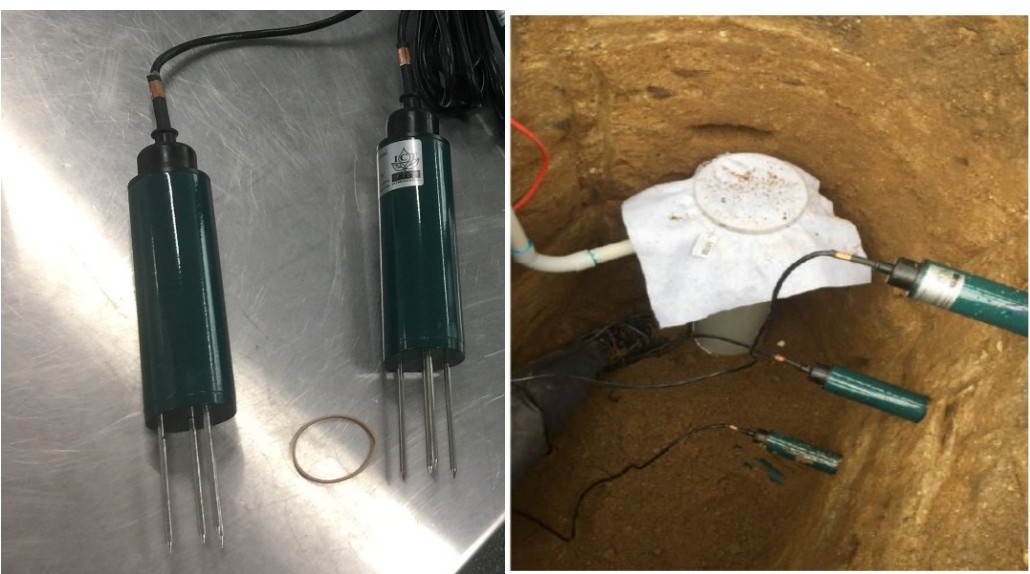

**Figure 4.** MP406 soil moisture sensor (**left** image) and sensors implanted in different depth of soil (**right** image) [37].

### 3.4.2. Leaf Wetness Sensor

Leaf wetness is a parameter which describes the existence and amount of water on leaf surfaces. The leaf of a plant often intercepts water during fog and dew or rain. Water can also be trapped in a leaf if the plant undergoes excessive irrigation [38,39]. Leaf wetness content provides an optimum benchmark for the moisture status of the entire plant. Knowledge of water status in plants provides farmers with a better perception of drought and its origin [40]. Furthermore, it allows agronomists to regulate the watering method by determining whether crops require water and the needed volume [39–41]. Leaf wetness or turgidity is an essential factor for the photosynthetic performance investigation. Thus, it can provide crucial information about the health of plants [40]. Excessive leaf turgidity often leads to the sporulation and occurrence of a range of fungal diseases that can perturb the plant growth. An appropriate understanding of the degree and duration of leaf turgidity allows agronomists to evaluate the right areas and time to adopt protective

measures such as deploying fungicides [39]. Leaf wetness sensing devices enable direct estimation of the hydration of plants. Therefore, they offer an effective and potential replacement of the indirect hydration measurement systems through soil moisture or temperature sensors (Figure 5). Such meticulous and self-sustaining devices can prevent plant damage and increase crop yields while conserving water resources [41].

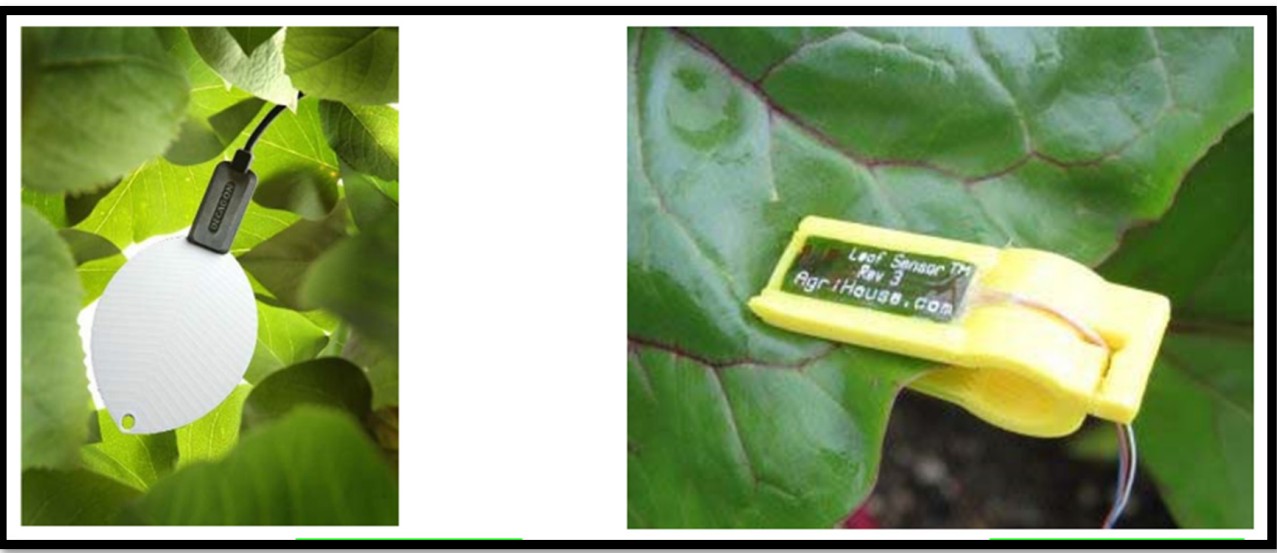

**Figure 5.** Decagon leaf wetness sensor (**left** image) and AgriHouse leaf sensor (**right** image) [41].

### 3.4.3. Soil Salinity Sensor

Soil salinity analysis is critical for crop yield in the agronomical sector. Salinity is the measure of soluble salts concentration in soil. Sodium chloride (NaCl) is the most common among all these salts; however, other types of salts include sulfates and carbonates of calcium, potassium, and magnesium. The growth of a particular species of plant requires a certain salinity level in the soil. The soil salinity level controls the osmosis process of water into the plant roots. However, soil with increased salinity often perturbs the growth of crops, pastures, and plants since it interferes with the nitrogen absorption. Such interference eventually causes plant dehydration [42,43]. In addition to the osmotic effect of salts in soil solution, excessive amount and absorption of individual ions can also become harmful to the plants. Consequently, it obstructs the plants from absorbing the other necessary nutrients. In turn, this prompts the need for a soil salinity monitoring device in the site-specific farming system [44]. To date, several initiatives have been originated for developing soil salinity measurement devices or sensors. However, the majority of such devices are either quite costly or have a highly complex system. Moreover, to carry out an effective measurement, these sensors often need an increased amount of soil solution [45]. We recommend a Radio frequency Identifier (RFID) based salinity sensor [41] as shown in Figure 6.

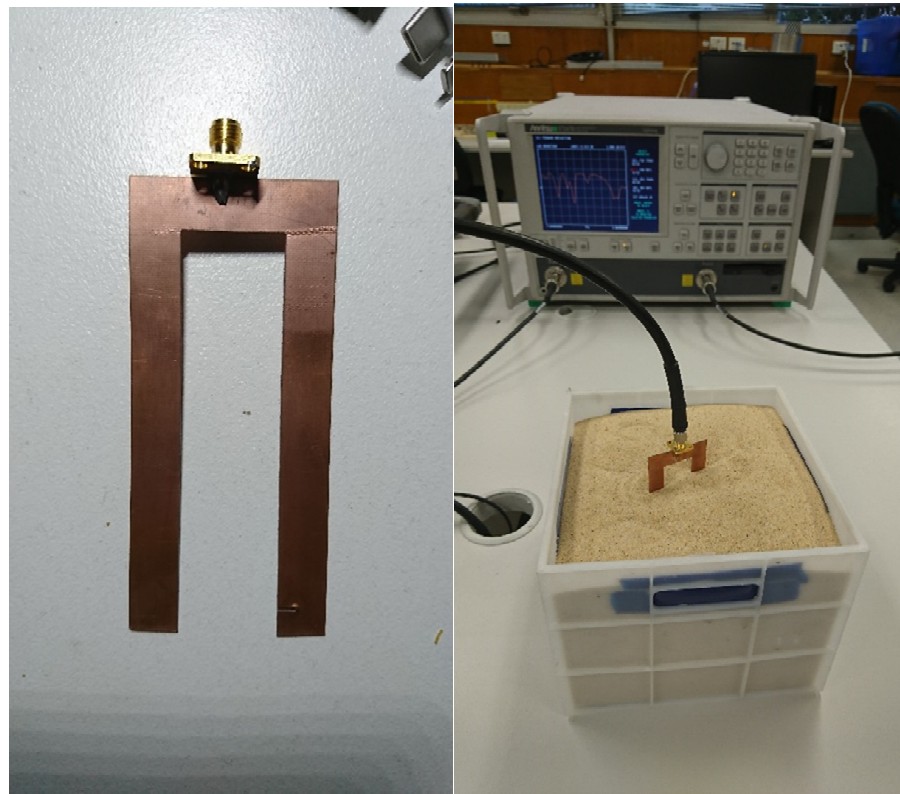

**Figure 6.** RFID based Salinity sensor (**left** image) and salinity sensor emerged in soil (**right** image) [42].

### 3.4.4. Soil Temperature Sensor

Soil temperature is also a regulatory feature in the agricultural sector. It is a critical catalyst in vegetation growth and soil biological activity. It has a significant impact on seed sprouting, root, and shoots growth, and nutrient absorption and crop production [46]. When the soil is cold, the seed may not germinate in the ground. Even if it does, it remains fragile and lacks the strength and vigor needed to develop appropriately and hence becomes vulnerable to succumbing to pests and disease. Soil temperature also influences soil moisture content, aeration, and availability of plant nutrients. The organic processes for nutrient availability and transformations are controlled by soil temperature. When soil temperatures are low, certain nutrients become unavailable or less available to plants. This is particularly true in the case of phosphorus, which largely promotes the development of roots and fruit in plants [47]. Soil microorganisms exhibit optimum upsurge and activity at an optimized temperature range. Essentially, most of the crop yield is slowed if the temperature is under approximately 900 °C and above 500 °C [48]. Therefore, it is essential to identify if the soil temperature is suitable to plant growth and microorganisms related activities. Some soil temperature sensors (Figure 7) are commercially available to address this issue.

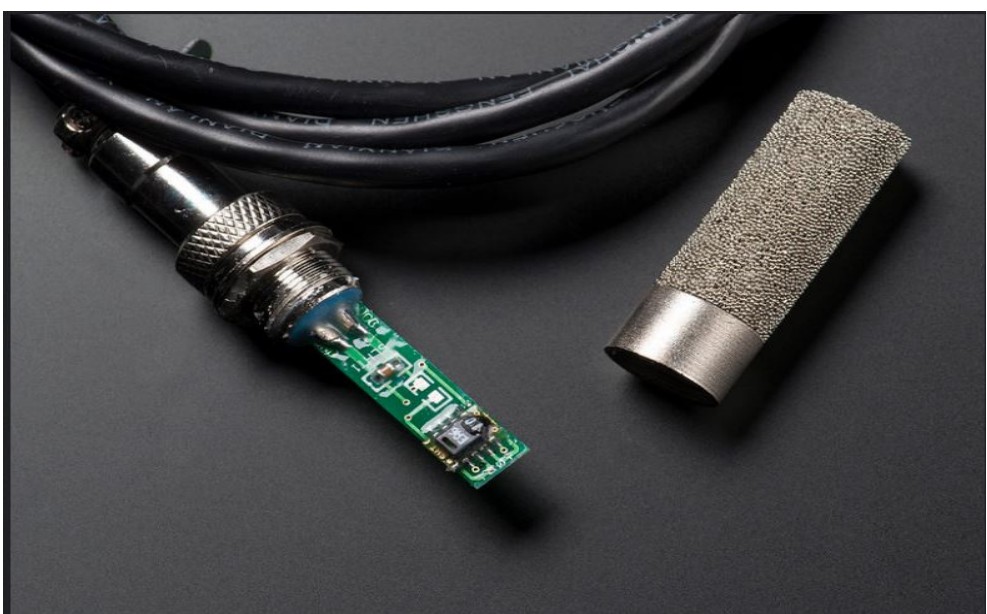

**Figure 7.** Soil temperature sensor [49].

This paper recommends paper-based Chip-less RFID [41] as one of the effective sensors and controls, which may provide an excellent opportunity to monitor smart farming. Accordingly, smarting farming can benefit from passive sensors with no battery, low cost, robust, compatible with harsh environments, and require no maintenance.

*3.5. Farm Management Information System (FMIS)*

Farm management information system (FMIS) is defined as a system that stores and processes farm-related data and supports automatic data acquisition, monitoring, planning, documenting, and decision making for farm management [50–52]. FMIS assists farmers in the execution and documentation of farm activities and helps them evaluate and optimize the data. They are also helpful in strategic, tactical, and operational planning of the farm operations. The evolutionary IoT based FMIS is expected to become part of the commercial FMIS, as they will cover different needs across the supply chain and the needs of IoT based agriculture. In addition, decision support systems (DSS) are essential in dealing with Big Data and assisting the farm manager in management and decision making in tasks such as farm financial analysis, business processes, or supply chain functions. Therefore, in this paper, we propose the AI and IoT based smart farming to support the farmers to have better control over their farms by monitoring, controlling, and forecasting the parameters affecting the crop health. Consequently, by improving the crop yields, further benefits can be achieved by AI and IoT based smart farming system in Bangladesh:

- Input efficiency—agricultural inputs (e.g., water, soil, pesticides, fertilizers, etc.) and its amount of application efficiency;
- Reduction of cost—it will reduce the production cost through the control and systematic application of the agricultural inputs;
- Profit—it will raise the profit by increasing the production rate of the crops per hectare;
- Climate—it will play a vital role in climate protection (e.g., reduce the amount of chemical use, reduce the emission due to the irrigation and other electric equipment service in the farm, etc.). Hence, AI and IoT based smart farming will help Bangladesh to reach the goal of sustainable agriculture to eliminate hunger by 2030 [53,54].

**4. Proposed AI and IoT Based Autonomous Smart Irrigation System in Bangladesh**

In addition to the above discussions, smart irrigation will play a significant role in smart farming in Bangladesh. The sustainability management system in agriculture de-

mands that water consumption must be reduced to preserve water resources. However, about 70% of the freshwater is being used to irrigate rice lands in Bangladesh. For example, it has been estimated that around 2.36 million hectares of land require artificial irrigation for Boro rice cultivation [53]. The reduction of groundwater level is an important issue to implement smart farming in Bangladesh. One of the primary reasons for this is the massive use of groundwater for Boro rice field irrigation, especially in the northern part of the country, including Rajshahi and Rangpur divisions. Hence, for the sake of sustainable agriculture, smart technologies are most desirable. Researchers proposed a practical deployment scheme of IoT based smart sprinklers using LoRaWAN protocols in remote smart farming [20]. Therefore, following the context of Bangladesh and previous studies [20,51–53], we propose an AI and IoT based smart irrigation system to smart agriculture so that irrigation can be done effectively.

In Figure 8, a smart irrigation system is proposed, where real-time weather data and real-time soil moisture sensor data can be collected and transmitted to servers using LoRaWAN gateway. Soil moisture sensors will collect the moisture level of soil, and the weather sensor will forecast the weather condition. When the soil moisture level goes below a predefined threshold level, and the weather forecast does not show any probability of rain, then AI will make a decision, and the sprinkler will be turned on. Consequently, resources such as water and electricity can be used efficiently, and the probability of overwatering or underwatering can be avoided. To serve this purpose, we propose to create an IoT network consisting of sensors (weather sensor and soil moisture sensor), LoRaWAN gateway, network server, and application server. Using the collected real-time data, an AI based algorithm can be used to make the optimal decision for controlling the automatic sprinkler.

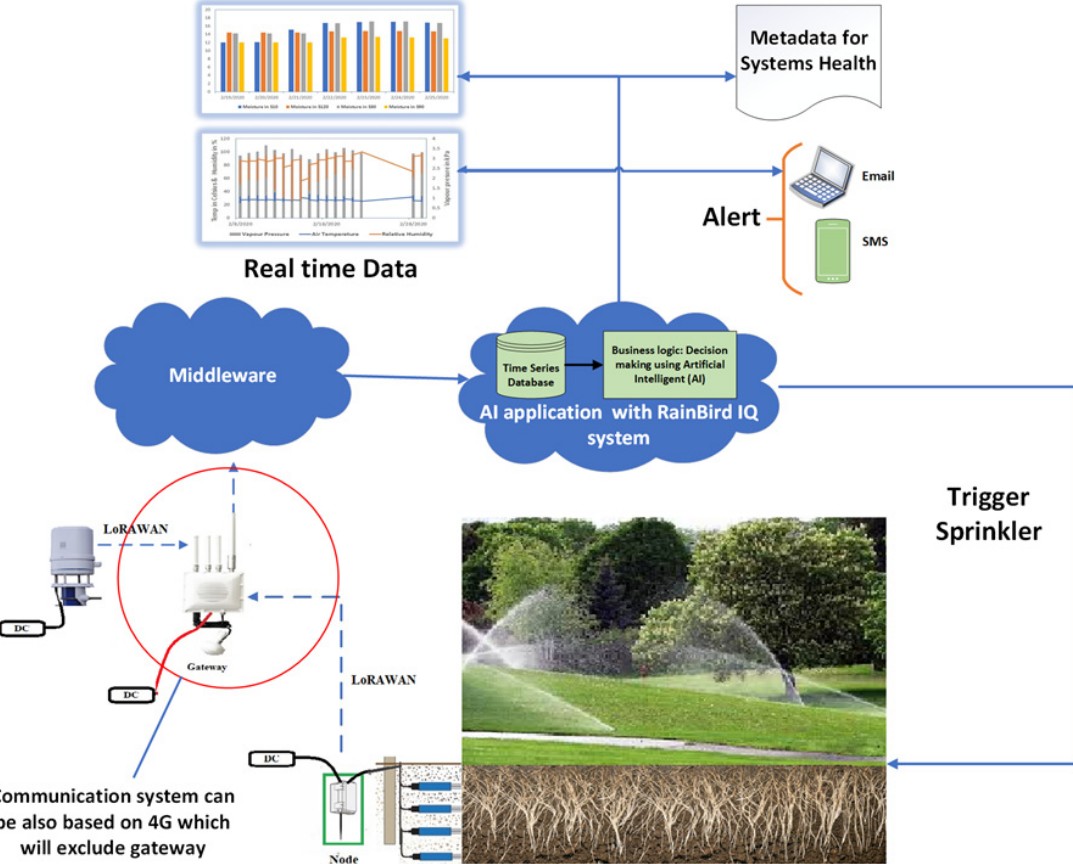

**Figure 8.** Proposed AI and IoT based smart irrigation system [37].

Extensive research shows that smart farming and leadership have underlying effects on sustainable agriculture [9,54]. Researchers suggested a roadmap with leadership to develop and enact smart farming [3]. They identified, for the multilevel leadership, inputs from government, NGOs, commercial sectors, and international funding bodies to achieve smart farming to enhance farmers' productivity and sustainable agriculture. Therefore, considering this proposed AI and IoT based smart farming, this paper calls for the demand for the responsible leadership to make it a reality for Bangladesh. Figure 9 depicts our proposed smart irrigation system to achieve smart farming in Bangladesh.

## 5. Responsible Leadership

The concept of responsible leadership refers to societal and interpersonal aspects that focus on the leader–follower relationship from the stakeholder perspective [10,11]. Scholars suggested that the belief of responsibility lacks from other leadership approaches that include value component, such as ethical, authentic, transformational, servant, and leadership [10,55]. Value-based leadership refers to leaders' conscious and moral obligations to manage their value components to create an environment that enhances economic performance, ethical concerns, and social engagement to overcome adverse environmental challenges [7,55]. Responsible leadership as a multilevel concept (e.g., individual, group, and organizational) evolved from various theoretical backgrounds such as stakeholder, agency, stewardship, institutional, and ethical theories [10,11,55]. It promotes the stakeholder theory [7] and combines the two significant study fields—social responsibility and leadership [55]. Hence, responsible leadership is recognized as a more comprehensive and inclusive leadership approach, far superior to any single value-based leadership theory such as transformational, charismatic, authentic, participative, servant, and ethical leadership. However, much has been explored about social responsibility, such as its relationship to organizations' financial performance outcome [7,55], less is understood about how activities and judgments on the part of individuals (e.g., microlevel for farmers) affects social responsibility (e.g., macrolevel for sustainable agriculture).

We recommend responsible leadership for smart farming and highlights sustainable agriculture. Responsible leadership from the microlevel includes the consideration of individuals' behavior and responses and their relations with leadership influences [56]. Scholars suggest that responsible leaders play a significant part in a sector as role models [10,11]. They involve their followers in decision-making for a greater good and mutual benefit. As a result, followers reflect a higher level of work satisfaction and motivation. Several scholars have advised that responsible leaders consider their followers as vital stakeholders to make use of their unique and diverse perspectives in maintaining their performance [10,11,56].

On the other hand, from the macrolevel perspective, responsible leaders can build an open, comprehensive, and varied societal culture by communicating and sharing knowledge while promoting robust ties with stakeholders that leads to higher social growth, innovation, and productivity [10,11,56]. Hence, responsible leadership goes beyond the traditional leader–follower to a leader–stakeholder relationship and advises that "building and cultivating . . . ethically sound relations toward different stakeholders is an important responsibility of leaders in an interconnected stakeholder society" (p. 101) [57]. Accordingly, we suggest that responsible leadership with smart farming will increase the productivity of Bangladeshi farmers and introduce a leader–stakeholder relationship among the policymakers and agricultural stakeholders to install sustainable and smart farming in Bangladesh.

## 6. The Role of Responsible Leadership for Smart Farming

Responsible leadership can contribute to smart farming in Bangladesh effectively compared to any other leadership approach because of the following major reasons. First, responsible leadership emerged to avoid leadership failures due to ethical and questions linked to socioeconomic expectations, which have not been considered in the previous

literature, particularly for the agricultural and sustainable social development [10,11,56]. Second, responsible leaders can manage the relationships among the people and their stakeholders in an inclusive manner that the outcomes become more ethically acceptable, socially desirable, and sustainable [56,57]. Third, responsible leadership may offer a systematic approach to identify, evaluate, and manage both the ethical and stakeholders' expectations to implement a socially sustainable smart farming system following Bangladesh's socioeconomic conditions.

The proposed conceptual model (Figure 1) suggests that smart farming enabling responsible leadership can influence and upgrade the use and impact of AI and IoT based technologies on sustainable agriculture. The enabling role of responsible leadership in this paper includes the following justifications.

First, smart farming enabling responsible leadership may strengthen outcomes such as sustainable agriculture. For example, increasing farmers' knowledge, skill, and ability regarding AI and IoT can enhance the performance and sustainability of their existing practice of traditional to smart farming [3,9]. Several scholars demonstrated positive links between leadership influence and smart farming [9]. Accordingly, current studies have also indicated higher agricultural sustainability through smart farming [9,54].

Second, the proposed model expects that the inclusion of responsible leadership in the agricultural sector will reinforce the positive outcomes for the AI and IoT based applications in the current farming practices in Bangladesh. This is because enabling responsible leadership into the farming sector and work ethic through smart farming will help both the policymakers and farmers to go beyond traditional farming applications and safeguard higher farming income. For example, FMIS, weather, and soil moisture sensors at farming practices will facilitate both the farmer and traditional cultivation practices to move more toward smart farming. Hence, the implication of responsible leadership into the agricultural sector to smart agriculture could increase sustainable agriculture [9,54]. Besides, combining responsible leadership with smart farming may increase the bonding and trust relationships among farmer and their support providers (e.g., local government agriculture development institutes, NGOs, or banks) for a higher level of agro-based income through agricultural sustainability [3]. We claim that the agricultural sector aiming smart farming toward enabling responsible leadership may increase the effective use of AI and IoT applications, which will significantly achieve agricultural sustainability (Figure 9).

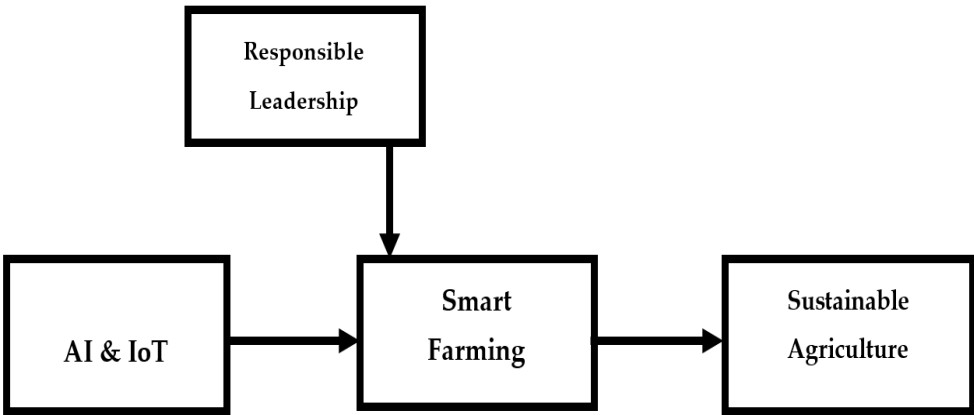

**Figure 9.** AI and IoT based sustainable smart farming enabling responsible leadership.

## 7. Directions for Future Research

We suggest that exploring the relationships among AI and IoT based smart farming, enabling responsible leadership and their microlevel and macrolevel outcomes have several implications. At present, there is a lack of knowledge and empirical evidence about the opportunity of AI and IoT based technologies and their prospect for smart farming in the context of developing countries. Hence, further delays in future studies may cost more for

the agriculture sector [6,58]. We suggest that research on AI and IoT based smart farming enabling responsible leadership should encompass earlier interventions. In addition to the primitive farming culture, Bangladesh is also experiencing adverse environmental and climate change impacts caused by intensive cultivation practices [6,58]. Hence, smart farming enabling responsible leadership will be significant for Bangladesh to exploit the opportunity of AI and IoT based smart farming establishment. For example, a new approach in agriculture, such as Agri-Food 4.0 [4] can be followed for quicker results for sustainable agriculture. As a result, the prospect of IoT, smart farming enabling responsible leadership in the agriculture sector will not only boost the national economy of Bangladesh but will also achieve SDG goals (SDG 2.4.1: Sustainable agriculture) [54]. The Government of Bangladesh has applied several strategies in this regard: for example, the National Adaptation Programs of Action (NAPA) in 2005, the Bangladesh Climate Change Strategy and Action Plan, and the 2016–2021 Five-Year Plan [12]. The Bangladesh government showed notable progress towards achieving the millennium development goals (MDGs) including higher agricultural productivity, alleviating poverty, reducing both infant and maternal mortality numbers, and reducing the occurrence of communicable diseases [59,60]. However, the government has not yet been able to make the best strategic initiative plan to apply the cutting-edge AI and IoT technologies for the agriculture sector. We suggest that the transformation of traditional farming to smart farming will require a comprehensive futuristic plan and investment, including responsible leadership training and development for all the policymakers, local government officials, and farmers. Referring to the NAPA or MDGs initiatives in Bangladesh, this paper predicts that the process of transforming traditional farming into smart farming, a period of 3 to 5 years may be needed to achieve agricultural sustainability.

Smart farming also features the concept of precision agriculture (PA) as information and technology-based system. It uses one or more of the following data sources: soils, crops, nutrients, moisture, or yield, for optimum profitability, sustainability, and protection of the environment [61]. It involves the observation, impact assessment, and timely strategic response to fine-scale variation in an agricultural production process [62]. For precision farming, it is essential to have accurate information regarding soil and crop factors' spatial and temporal variability within a field [63,64]. Therefore, it uses sensor technologies for yield mapping and prediction, soil sensing, irrigation control, etc., to address the site-specific needs with spatially variable application [62,64]. Previous studies on the agricultural sector in Bangladesh have mainly focused on the production and environment analysis [6,58]. However, differences might exist between the intended practices at the national (or macro) level and the implemented practices and farmers' (or microlevel) perceptions. Therefore, it is essential to include multilevel research at the individual or farmer level, and on research at the policymakers' level. Finally, we suggest that future researchers may explore multilevel and multimethod, such as specific agricultural items and longitudinal study (e.g., 12–18 months) and multisource data collection.

## 8. Conclusions

Given the increased interest and prospect of AI and IoT in smart farming and responsible leadership, more theoretical development is necessary for the literature. We addressed several research issues related to AI and IoT based smart farming and responsible leadership in proposing Figure 9. We argued that AI and IoT based smart farming have a significant impact on sustainable agriculture and responsible leadership may play a crucial role in offering mutual benefits for stakeholders, including farmers, societies, environments, and the national and global economy as a whole. The overall underlying idea is that AI and IoT based technology, combined with smart farming enabling responsible leadership, will ensure sustainable agriculture in Bangladesh. The challenge is to go beyond the simple application of traditional agriculture practice and leadership in the agriculture sector and develop a continuous and sustainable climate. We expect that our proposed AI and IoT

based smart farming enabling responsible leadership will achieve and foster smart farming with future innovation and efficiency.

**Author Contributions:** Conceptualization, A.H., N.I., N.H.S., S.D. and B.R.; methodology, A.H., N.I., N.H.S., S.D. and B.R.; software, A.H., N.I., N.H.S., S.D. and B.R.; validation, A.H., N.I., N.H.S., S.D. and B.R.; formal analysis, A.H., N.I., N.H.S., S.D. and B.R.; writing—original draft preparation, A.H., N.I., N.H.S. and S.D.; writing—review and editing, A.H., N.I., N.H.S., S.D. and B.R.; visualization, A.H., N.I., N.H.S., S.D. and B.R.; supervision, A.H., N.I., N.H.S., S.D. and B.R.; project administration, A.H., N.I., N.H.S., S.D. and B.R. All authors have read and agreed to the published version of the manuscript.

**Funding:** This research received no external funding.

**Conflicts of Interest:** The authors declare no conflict of interest.

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
