# Peer review of "Smart Farming through Responsible Leadership in Bangladesh: Possibilities, Opportunities, and Beyond"

_sustainability, doi:10.3390/su13084511_

Round 1
Reviewer 1 Report
This study discussed the potential and opportunities of smart farming w/ responsible leadership in Bangladesh. The authors did a nice review on smart farming, but improvements can be made when linking it w/ responsible leadership. I suggest the authors add more discussion on to what extent responsible leadership can contribute to smart farming. In current manuscript, these sections feel weak.
Also, please double-check that you used the correct references. For example, I couldn't find the text indicating the economic loss of 33 billion $ by pest (line 320-321) in ref 31 (additionally, the link of Ref 31 is pointed to another paper).
Author Response
Please see the attached here.

Reviewer 2 Report
It's time to think how to prepare the world for 2050, in the aspect of agriculture development concerning food production for actual population. This paper discuss the opportunities in challenging world taking into acount leadership centers, which could utilize 5 G wireless technology, Agri-Food 4,0 of digitalised productivity, as well efficient land management to achieve agricultural sustainability. The responsible leadership in agriculture activity can provide quite high possibilities in obtaining right quality of agriculture production, which can be spread in developing countries, as good agriculture practice. All these is very important esspecially in time of climate change periode.
I would suggest to introduce short discussion about Precision Agriculture, which is very close to Smart Farming and Artificial Intelligence, esspecially if we consider Agriculture Technology.
Presented in the paper detailed description of special sensors and controls give great opportunity to provide monitoring of different agriculture processes, but authors could give some examples of such developments.
Author Response
Please see the attached here.
